# Multi-Omics Approach to Mitochondrial DNA Damage in Human Muscle Fibers

**DOI:** 10.3390/ijms222011080

**Published:** 2021-10-14

**Authors:** Matthias Elstner, Konrad Olszewski, Holger Prokisch, Thomas Klopstock, Marta Murgia

**Affiliations:** 1Department of Neurology, Technical University Munich, 81675 Munich, Germany; m.elstner@tum.de; 2Center for Addictive Disorders, Department of Psychiatry, Psychotherapy and Psychosomatics, Psychiatric Hospital, University of Zurich, 8001 Zurich, Switzerland; kolsz@gmx.net; 3Institute of Human Genetics, Technical University Munich, 81675 Munich, Germany; prokisch@helmholtz-muenchen.de; 4Institute of Neurogenomics, Helmholtz Zentrum Munich, 85764 Neuherberg, Germany; 5Department of Neurology, Friedrich-Baur-Institute, University of Munich, 80336 Munich, Germany; Thomas.Klopstock@med.uni-muenchen.de; 6German Center for Neurodegenerative Diseases (DZNE), 81675 Munich, Germany; 7Munich Cluster for Systems Neurology (SyNergy), 81675 Munich, Germany; 8Department of Proteomics a Signal Transduction, Max Planck Institute of Biochemistry, 82352 Martinsried, Germany; 9Department of Biomedical Sciences, University of Padova, 35131 Padua, Italy

**Keywords:** skeletal muscle, mtDNA deletions, transcriptomics, proteomics, myopathy, disease models

## Abstract

Mitochondrial DNA deletions affect energy metabolism at tissue-specific and cell-specific threshold levels, but the pathophysiological mechanisms determining cell fate remain poorly understood. Chronic progressive external ophthalmoplegia (CPEO) is caused by mtDNA deletions and characterized by a mosaic distribution of muscle fibers with defective cytochrome oxidase (COX) activity, interspersed among fibers with retained functional respiratory chain. We used diagnostic histochemistry to distinguish COX-negative from COX-positive fibers in nine muscle biopsies from CPEO patients and performed laser capture microdissection (LCM) coupled to genome-wide gene expression analysis. To gain molecular insight into the pathogenesis, we applied network and pathway analysis to highlight molecular differences of the COX-positive and COX-negative fiber transcriptome. We then integrated our results with proteomics data that we previously obtained comparing COX-positive and COX-negative fiber sections from three other patients. By virtue of the combination of LCM and a multi-omics approach, we here provide a comprehensive resource to tackle the pathogenic changes leading to progressive respiratory chain deficiency and disease in mitochondrial deletion syndromes. Our data show that COX-negative fibers upregulate transcripts involved in translational elongation and protein synthesis. Furthermore, based on functional annotation analysis, we find that mitochondrial transcripts are the most enriched among those with significantly different expression between COX-positive and COX-negative fibers, indicating that our unbiased large-scale approach resolves the core of the pathogenic changes. Further enrichments include transcripts encoding LIM domain proteins, ubiquitin ligases, proteins involved in RNA turnover, and, interestingly, cell cycle arrest and cell death. These pathways may thus have a functional association to the molecular pathogenesis of the disease. Overall, the transcriptome and proteome show a low degree of correlation in CPEO patients, suggesting a relevant contribution of post-transcriptional mechanisms in shaping this disease phenotype.

## 1. Introduction

The clinical spectrum of mitochondrial DNA (mtDNA) deletion syndromes is highly variable, ranging from mild myopathy to the severe syndromic Kearns-Sayre Syndrome (KSS) [1]. The most frequent adult presentation is chronic progressive external ophthalmoplegia (CPEO), a generalized myopathy that typically manifests with ptosis and ophthalmoparesis [2]. Single size mtDNA deletions generally occur sporadically and are in most instances not transmitted to offspring. In contrast, multiple deletions are caused by mutations of nuclear genes coding for the mitochondrial replication machinery and thus show Mendelian transmission [3,4]. Within a single cell or tissue, mutated mtDNA can coexist with wild type molecules, a situation referred to as heteroplasmy (expressed in %). The clinical heterogeneity is considered to depend on the inter- and intraindividual degree of heteroplasmy and size and location of the deletions, but also nuclear disease modifiers and environmental factors [5,6,7]. Furthermore, mtDNA deletions also occur in muscle fibers and neurons as a function of age [8].

Diagnostic workup of patients includes histochemical analysis of skeletal muscle biopsies and typically shows cytochrome c oxidase (COX; complex IV of the respiratory chain) deficient fibers (COX-) next to fibers with normal COX activity (COX+). The 13 subunits of COX are encoded part nuclear (10 subunits), part mitochondrial (3 subunits) and mtDNA deletions affect COX subunit expression and function in relation to heteroplasmy levels [9]. In COX-negative fibers, the successive histochemical probing for succinate-dehydrogenase (SDH; complex II) reveals normal or increased activity of this solely nuclear encoded enzyme, resulting in a typical brown/blue mosaic pattern (see Figure 1A) [10].

Heteroplasmy levels >30% can cause a decline in basic bioenergetic parameters, such as ATP synthesis [11]. This initially is compensated by induction of mitochondrial biogenesis, histologically leading to ‘ragged red fibers’ (RRF) [12]. Furthermore, mtDNA deletions are reported to cause a bioenergetic adaption by activation of autophagy and salvage pathways through AMPK and mTOR pathway regulation [13]. Failure to maintain mitochondrial function progresses to decreased energy production and increased generation of reactive oxygen species (ROS). In skeletal muscle this leads to weakness, fatigue, and exercise intolerance [1]. Using mass spectrometry-based proteomics, we have previously shown that COX- fibers increase the expression of TCA cycle and fatty acid beta-oxidation enzymes [14]. Other studies have shown that increased production of ROS correlates with OXPHOS deficiency, confirming the link between mitochondrial dysfunction and oxidative damage of DNA, proteins, and lipids [15]. The phenotypic threshold effect may differ greatly for in vivo patient tissue and model systems, such as cybrid cell lines, but for mtDNA deletions heteroplasmy levels >60% are frequently reported to be deleterious [16,17]. Recent findings support the notion that heteroplasmy in the maternal germ line is influenced by the nuclear genetic background, which in turn shapes the characteristics of mtDNA in the human population [18].

In this study we used whole genome transcriptome analysis, to characterize the molecular phenotype of COX-deficient fibers and COX-positive fibers, isolated from the same patient using laser microdissection. The transcriptome consists of all RNA molecules within a cell, the majority of which are non-coding. In this study, we have focused our analysis specifically on messenger RNAs (mRNAs). The expressed transcriptome, in turn, is the template for the cellular proteome, the repertoire of proteins found in COX+ and COX− muscle fibers. Previous transcriptome studies have measured a downregulation of transcripts encoding proteasome activity, and a concomitant induction of transcripts linked to the AMP kinase pathway, autophagy, and amino acid degradation as an adaptation to the bioenergetic deficit caused by mtDNA [13,19]. This intriguingly complex network converges on the mTOR pathway which, in turn, mediates the stimulation of PPAR/PGC-1α and induces mitochondrial proliferation [20]. Proteomic studies of various mitochondrial disorders have shown that the muscle of patients with COX deficiency upregulates mitochondrial biogenesis and redirects the metabolic substrate flux from the respiratory chain to tricarboxylic acid cycle (TCA) in response to respiratory chain deficiency [21]. The mechanisms leading to adaptive responses as well as cell fate and demise are diverse and not fully understood. Adaptation to stress leading to bioenergetic deficits includes complex variations in mitochondrial dynamics. In particular, energy depletion has been shown to cause elongation of the mitochondria, which prevents their degradation and preserves cell viability [22].

With the aim of exploring the cellular adaptations to bioenergetic deficit caused by mtDNA mutations using omics technologies, in this work we have directly compared the transcriptome of COX+ and COX− fibers isolated by LCM from muscle sections. While our approach allows us to directly compare functioning and bioenergetically deficient fibers within the muscle of the same patients, several omics studies mentioned above were based on total lysate of muscle biopsies and compared subjects carrying different pathological mutations [21] or myoblasts derived from patients compared with those of controls [23]. Omics technologies allow us to derive large-scale network analyses of genes with significantly different expression in COX-deficient fibers and COX-positive fibers. As we had previously carried out a proteomic study comparing COX+ and COX− fibers from CPEO patients, we have here crossed the information obtained with the two different omics techniques on patients with the same mitochondrial disorder. We found a general lack of correspondence between the transcriptome and proteome data in detailing the phenotype of COX deficiency. This finding highlights the role of post-transcriptional mechanisms in controlling the phenotypic adaptations of the muscle fiber proteome in mitochondrial diseases. 

## 2. Results

### 2.1. Workflow of Histochemical COX/SDH Staining Coupled to Laser Capture Microdissection and Omics Analysis

We carried out a COX/SDH staining in cross sections of muscle biopsies from nine CPEO patients (Appendix A) Using the images acquired in stained sections as a reference, we then identified the corresponding COX+ and COX− unstained fibers in serial sections and dissected them by LCM using a laser beam with a wavelength of 349 nm. Individual fiber sections corresponding to the defined regions of interest were catapulted into the caps of 0.5 mL capture tubes (Figure 1A). While some patients displayed increased central myofiber nuclei, indicative of muscle regeneration, nuclei were mostly in typical subsarcolemmal position for most subjects (Appendix A). From the biopsy of the same patient, pools of 500 COX+ and COX-sections were collected and immediately frozen in liquid nitrogen. The procedure was repeated for all biopsies of nine patients. RNA was isolated and in vitro transcribed for hybridization on Illumina expression chips. In parallel, we carried out a meta-analysis of the proteomics dataset that we had previously generated using the same histochemical COX/SDH coupled to LCM in three of CPEO patients [14]. In this case, we had analyzed the samples by liquid chromatography coupled to tandem mass spectrometry (schematically represented in Figure 1B).

We integrated the transcriptome and proteome datasets by matching the gene products based on gene name/target ID, taking a protein-centric view [24]. This procedure yielded a combined table of over 12,000 quantified gene products, 3904 of which could be univocally matched in both datasets (Figure 1C). The transcripts for which we could not find a corresponding protein were enriched in tRNA and in transcription factor-binding and transcription factors annotations. The latter are low abundance proteins underrepresented in the fiber proteome. Therefore, their sequencing and quantification by mass spectrometry-based proteomic is often hindered by highly abundant sarcomeric proteins determining a high dynamic range [25]. Likewise, many of the proteins for which we could not map a corresponding RNA are plasma and mitochondrial proteins, suggesting that the corresponding RNA may be underrepresented in the muscle fiber transcriptome (Figure 1D).

### 2.2. Comparative Transcriptome Analysis of COX+ and COX- Fibers 

The progressive loss of oxidative capacity in a subset of muscle fibers from CPEO patients is due to increasing heteroplasmy for the mtDNA mutation, reaching a threshold that irreversibly impairs the expression and assembly of the respiratory chain. We measured an average 34% mtDNA heteroplasmy in COX+ and of 60% in COX- fibers, confirming that the phenotypic characterization obtained by histochemical staining matches the molecular analysis of the mitochondrial genotype (Figure 2A). The phenotypic difference between COX+ and COX- fibers was evident at the whole transcriptome level. We applied principal component analysis to the whole dataset of 9247 RNAs quantified in CPEO muscle fibers of nine subjects. A net diagonal separation of COX+ and COX- fibers along components 2 and 4 could be observed for both male and female subjects (Figure 2B). While the drivers of the separation of COX+ fibers were rather heterogeneous comparing males (top) and females (bottom), many common hits were driving the separation of COX- fibers. This suggests that the transcriptomic features of the OXPHOS deficiency in CPEO are largely gender independent (Figure 2C).

We set out to further investigate the effects of OXPHOS deficiency by carrying out a paired two-sample Student’s *t*-test between COX+ and COX- fibers. Using Benjamini-Hochberg false discovery rate (FDR) for cut-off, we obtained a list of 148 gene products displaying significant expression differences between the two groups (Appendix A). Unsupervised hierarchical clustering carried out on the median expression in male and females yielded two groups of transcripts with a opposite pattern of expression in COX+ and COX- fibers with only minor differences between genders (Figure 2D and insets in yellow). We looked for specific annotation enrichments within the two groups of transcripts with differential expression in COX+ and COX- fibers. Fisher’s exact test applied to transcripts with higher expression in COX+, using a human expressed gene list of 22,000 hits as a background, yielded as highest significant enrichments muscle cytoskeleton and ubiquitin ligation. The same procedure in COX- highlighted ribosomal and translational elongation, as well as mRNA decay (Figure 2E). These latter findings suggest that increased transcription in COX- fibers might be one of the putative compensatory mechanisms used to counteract the energy deficit associated with the pathology.

### 2.3. Network Analysis of Differentially Expressed Transcripts in COX+ and COX- Fibers

We looked for annotation clusters with significantly different expression between COX+ and COX- fibers using the “Functional annotation cluster” tool of the DAVID Bioinformatics Resources [26]. This analysis retrieved seven clusters with at least one significantly enriched annotation (*p* < 0.05). Two of them contained overlapping mitochondrial annotations and were merged, yielding six functional annotation clusters with enrichment >1. The two most significant annotations of each cluster, marked by a bubble whose size is proportional to the number of annotations it contains, are listed on the right (Figure 3A). Beside mitochondrial, cytoskeletal (LIM domain), and ribosomal (rRNA processing) terms, we found that negative control of apoptosis and cell cycle inhibition were also enriched. The latter, together with the observed enrichment in oxidative stress terms, correlates with pathological features of mitochondrial disorders [27].

To understand how this pool of differentially regulated transcripts might contribute to the pathological phenotype observed in CPEO, we explored the functions of the corresponding proteins. To this aim, we used the STRING database tools to perform physical and functional interaction network analysis of the protein whose coding transcripts were differentially expressed between COX+ and COX- fibers [28]. After excluding proteins without direct connections, we filtered for functional interactions based on experiments, databases, and text mining and connected the nodes through lines whose thickness indicates the strength of functional data support. We then defined five clusters, corresponding to groups of proteins with tight functional interaction involved in fundamental features of skeletal muscle physiology (Figure 3B). Altogether, this system view of the proteins encoded by transcripts differentially regulated between COX+ and COX- aims at reducing the complex molecular landscape of mitochondrial disease to a discrete number of functional relationships. This resource will provide a basis for novel mechanistic hypotheses, thereby contributing to dissect the molecular pathogenesis of CPEO.

### 2.4. Integration of Transcriptomic and Proteomic Analysis of COX Deficiency

We merged our present data comparing COX+ and COX- fibers at the RNA level to a proteomics dataset with the same experimental design, which we had previously measured [14]. Data integration based on gene names (target ID) yielded 3904 single-identifier gene products which were quantified in both datasets. Using the common gene products list to compare individual samples within datasets, both transcriptome and proteome data showed high Pearson correlations (0.95 and 0.90 respectively), indicating robustness and stability of both workflows (Figure 4A). Between COX+ and COX- fibers of the same donor we could consistently calculate correlations higher than 0.90 (Figure 4A insets). However, the Pearson correlation between proteome and transcriptome samples was consistently very poor (0.31 on average). If transcription was the only regulator, the Pearson correlation of these two datasets should be close to one, considering minimal variations due to less-than-perfect technical reproducibility and respective dynamic range and quantification methods. Conversely, our finding suggests that events other than transcription, presumably post transcriptional mechanisms involving protein stability and turnover, play a major role in shaping the mitochondrial phenotype of adult muscle fibers.

We then asked whether the differences between COX+ and COX- fibers were similar within the transcriptome and proteome datasets. To highlight the functional classes of gene products with significantly different expression correlating with COX deficiency, we carried oud 2D annotation enrichment analysis [24]. We used a protein-centric approach for functional annotations, using Gene Ontology (GO), Kyoto Encyclopedia of Genes and Genomes (KEGG), and Uniprot Keywords. We then compared COX+ and COX- fibers at both the transcriptome and proteome level. Among the annotations significantly enriched in the transcriptome data set were those related to muscle contraction and sarcomeres. Their position in the graph, close to the middle but shifted to the left, shows that these annotations have slightly higher relative enrichment in COX+, suggesting subtle differences in the transcriptional control of these structural genes. Several annotations related to aminoacid metabolism, asparagine, leucine, and serine in particular, had higher expression in COX- fibers (Figure 4B). From this analysis at the RNA level no clear mechanistic link was apparent with the known phenotype of the disease. In particular we could not highlight differences in functional gene classes related to the mitochondrion. Conversely, the comparison of COX+ and COX fibers at the proteome level highlighted the respiratory chain, electron transport, and mitochondria in the left region of the graph, indicating high relative enrichment in COX+ fibers (Figure 4C). This readout confirms our previous analysis at the proteome level, showing significantly higher expression of respiratory chain components in fibers with intact COX activity compared with deficient ones [14].

To directly compare the differences between COX+ and COX-fibers, we calculated the COX+/COX- ratio (by subtraction of Log2 values) for the median value of both proteome and transcriptome datasets. We then represented their comparison as a scatter plot, whereby gene products with similar expression in COX-deficient and positive fibers (the vast majority) occupy a central area around 0 values (ratio = 1, Figure 4E). This procedure highlighted which gene products are higher in COX+ than in COX- fibers in both datasets (quadrant top right, top enriched annotation muscle protein), at the proteome level (quadrant top left), top enriched annotation respiratory chain and at the RNA level (quadrant bottom right, top enriched annotation mitocarta). This analysis confirms that COX+ fibers have a higher expression, both at the RNA and protein level, of sarcomeric and calcium binding gene products involved in muscle contraction (among which are TNNI and TPM3, see top right). This finding is likely correlated with the progressive muscle weakness characterizing disorders caused by mtDNA mutations [29]. From the mechanistic point of view, this finding might be linked to the profound alterations in excitation-contraction coupling and aberrant calcium homeostasis detected in various mouse models of mitochondrial disease [30]. Conversely, only the proteome dataset displays a higher expression of the respiratory chain in COX+ fibers (NDUFS2, NDUFA12 among others, top left). The relative abundance of mitochondrial gene products (annotation mitocarta, bottom right) was higher in COX+fibers.

To confirm the different behavior of proteome and transcriptome, we detailed the expression of two gene products from the upper left and lower right quadrants of the plot (Figure 4, arrows). NDUFA12, an accessory subunit of complex I, had a similar RNA expression in COX+ and COX- fibers. However, COX+ fibers expressed eight-fold more of the protein than COX- (Figure 4E top). EEF1G, a translation elongation factor, displayed the same protein expression level in COX+ and COX- fibers, but the level of its transcript was double in COX+. Taken together, these differences indicate that transcriptome and proteome contribute to the different phenotype of COX+ and COX- through largely distinct molecular mechanisms. In particular, the higher expression of OXPHOS gene products in COX+ fiber is likely under the control of post-transcriptional mechanisms.

## 3. Discussion

Mitochondrial DNA (mtDNA) mutations and deletions cause inherited mitochondrial disorders occurring with a variety of symptoms, essentially in any organ and at any age. Of broader significance, mtDNA mutations and deletions arising in post-mitotic tissue are implicated in several age-related processes, such as atherosclerosis [31], sarcopenia [32], as well as neurodegenerative disorders, e.g., Parkinson’s disease [33,34,35]. 

Mitochondria are the main source of reactive oxygen species (ROS) and mtDNA deletions are believed to occur during repair of oxidatively damaged mtDNA molecules [36]. Primary pathophysiological events following accumulation of mtDNA deletions include a decline in mitochondrial bioenergetics and increased production of reactive oxygen species, which—in a vicious cycle—may reinforce the bioenergetic defect through oxidative damage to DNA, proteins, and lipids [37,38]. Tissue-specific threshold levels determine the heteroplasmy that is sufficient to cause a biochemical deficit and compromise cell function [39]. As result, a bioenergetic rescue program is initiated, which includes the inhibition of proteasomal activity and stimulation of an autophagic transcript [13]. In muscle, mitochondrial proliferation is induced, which results in accumulation of mitochondria below the plasma membrane, leading to the appearance of the phenotypically characteristic ragged-red fibers. When compensatory mechanisms fail, muscle fibers are believed to undergo atrophy and death [40].

In this study we have used a system approach based on omics technologies to highlight pathogenic features and compensatory mechanisms that accompany the loss of respiratory chain function in mitochondrial disorders. In particular we have analyzed the mosaic muscle phenotype of chronic progressive external ophthalmoplegia (CPEO) patients, where muscle fibers with a functioning respiratory chain can be found next to various proportions of fibers with OXPHOS deficiency within the same muscle. By comparing the transcriptional profiles of COX-+ and COX- fibers obtained by laser microdissection, we identified 148 significantly regulated genes. Enrichment analysis showed that COX+ and COX- fibers differed significantly in the regulation of amino acid metabolism, cell death and cell cycle control. Over 60% of the proteins corresponding to the groups of differentially expressed transcripts had at least one functional interaction within another protein of this list, giving rise to an interconnected network. Through this system view, we could highlight a hub of proteins involved in cell cycle and cell growth, with functional links to translation and metabolism as well as muscle contraction. These findings complement the contributions of our previous studies linking PI3K/Akt, a fundamental controller of skeletal muscle growth, to mitochondrial disease [5]. In addition, our newly generated transcriptome dataset contributes a resource for the community, to mine novel molecular links between the phenotype of mitochondrial disease, i.e., the pathognomonic COX mosaicism, and the events leading to cell pathology. The data integration with our proteomic dataset, previously generated with the same laser capture microdissection technique and experimental design, adds to this resource the unique possibility of generating hypotheses that account for the relative contribution of transcriptional and post-transcriptional mechanisms. Indeed, our analysis showing a low correlation between the transcriptome and proteome in COX deficiency indicates that multidisciplinary approaches might be the key to the complex pleiotropic disease presentation of mitochondrial disorders.

## 4. Materials and Methods

### 4.1. Patient Characteristics

Muscle biopsies were collected according to a clinical protocol approved by the Ethics Committee of the LMU Munich (N. 198-15). Informed consent was obtained from all subjects involved in the study. Nine patients with biochemically proven mtDNA deletion syndromes were included in the transcriptome study (female = 5, male = 4) and three, (female = 1, male = 2) in the proteome study without overlapping between the two cohorts. Biopsies were taken from biceps brachii, triceps, deltoid, and vastus lateralis. Routine histochemical diagnostics of biopsies had shown COX-negative fibers and further investigation of muscle homogenate revealed the presence of mtDNA deletions. Most patients had single deletions (sd) with no family history, indicating sporadic occurrence (*n* = 6). When multiple deletions were seen, candidate genes were analyzed for mutations (*n* = 3) and in one patient mutations in the polymerase γ gene (POLG1) were proven. The patient cohort is detailed in Appendix A.

### 4.2. Cytochrome-C (COX)/Succinate Dehydrogenase (SDH) and Myosin Staining 

For histochemical staining (COX/SDH), muscle cross sections on glass slides were incubated in COX medium (100 μM cytochrome c, 4 mM diaminobenzidine tetrahydrochloride, and 20 μg/mL catalase in 0.2 M phosphate buffer, pH 7.0) for 90 min at 37 °C. Sections were then washed in standard PBS, pH 7.4 (2 × 5 min), and incubated in SDH medium (130 mM sodium succinate, 200 μM phenazine methosulphate, 1 mM sodium azide, 1.5 mM nitro blue tetrazolium in 0.2 M phosphate buffer, pH 7.0) for 120 min at 37 °C. Finally, they were washed in PBS, pH 7.4 (2 × 5 min), rinsed in distilled water, and dehydrated in an increasing ethanol series up to 100%, prior to incubation in xylene and mounting in Eukitt (Merck, Burlington, MA, USA). For skeletal muscle fiber-type identification, antibodies against different myosin heavy chain isoforms were used. Anti-myosin heavy chain ‘slow’ antibody was used to identify the MyH7 isoform in type I fibers (Merck, Burlington, MA, USA)). For type IIa fibers, we used myosin A4.74 antibody (obtained from the Developmental Studies Hybridoma Bank, developed by Helen M. Blau, The University of Iowa). For sequential staining we used a diaminobenzidine staining kit as well as an alkaline phosphatase staining kit (both from Abcam, Cambridge, UK). 

### 4.3. Laser Microdissection, RNA and DNA Isolation, RNA Quality Control

Approximately 500 sections of individually isolated COX-negative and COX-positive muscle fibers were pooled. RNA was extracted on the day of IVT, using AllPrep DNA/RNA Micro-Kit (Qiagen, Venlo, Netherlands) following the manufacturer’s protocol. RNA was eluted in 10 µL RNAse-free water and used directly for IVT without prior storage. We determined RNA integrity numbers (RIN) for RNA isolated from unstained and stained sections (Agilent 2100 Bioanalyzer, Santa Clara, CA, USA). 

### 4.4. DNA Analysis

Heteroplasmy (level of mutated mtDNA) and copy number (total amount of mtRNA molecules) were determined on whole muscle tissue by qRT-PCR as previously described [5]. Long-range PCR was performed according to Bender et al. [41].

### 4.5. In Vitro Transcription 

Ambion MessageAmpII**^®^** was used following the manufacturers recommendations (Thermo Fisher, Waltham, MA, USA). After the first round, aRNA was eluted in 100 µL RNAse-free H2O and the volume was reduced to 10 µL in a vacuum centrifuge. The second round IVT was performed using the Illumina^®^ TotalPrep™ RNA Amplification Kit (Thermo Fisher, Waltham, MA, USA). The second round of IVT yielded >3 µg cRNA with an average length of 800 bp and was used for hybridization on Illumina**^®^** WG6v1 expression chips.

### 4.6. Hybridization Procedures, Scanning and Data Acquisition

Standard hybridization was performed according to the manufacturer’s recommendations. The Illumina**^®^** BeadArray Reader (Illumina, San Diego, CA, USA) was used to scan the surface of beadarrays. The digital images from the intensity signals were stored as .tiff files and the corresponding signal intensity values were decoded using the Decode map files provided with each beadarray and converted into the Illumina**^®^** propriety format .idat files. After scanned microarray images of Sentrix**^®^** BeadChips collected from the Illumina**^®^** BeadArray Reader, image data files (.idat) were directly downloaded into BeadStudio Data Analysis Software (Illumina®,San Diego, CA, USA) for data visualization and analysis.

### 4.7. Immunohistochemistry

Mitochondrial respiratory chain activity was detected using a DAB staining kit. As primary antibodies MS105, MS204, and MS404, obtained from MitoSciences, were used for depiction of the mitochondrial complexes I, II, and IV, respectively [42]. 

### 4.8. Bioinformatic and Statistical Analysis

The Perseus software (v.1.6.14.0), part of the MaxQuant environment [43], was used for data analysis and statistics. Categorical annotations were provided in the form of UniProt Keywords, KEGG, and Gene Ontology (GOBP, GOCC, GOMF). Illumina expression units were used for transcript quantification and label free quantification (MaxLFQ) was used for protein quantification in the proteome samples [44]. For pools of COX+ and COX- sections from individual patients, 100 sections were pooled. Sample number was nine patients for transcriptome (*n* = 9 biological replicates) and three patients for proteomics (3 biological replicates, each in technical replicates). The proteome dataset contains missing values (NaN, not a number) for some proteins. Comparison of quantitative parameters between the two groups was performed using Student’s *t*-test. Threshold was determined by permutation-based false discovery rate, with *p* value below 0.05 considered significant. PCA and cluster analysis was performed in the Perseus software using logarithmic expression values. For hierarchical clustering, values were Z-scored and clustered using Euclidean distance for column and row clustering. 

### 4.9. Data Availability

The transcriptomic data generated for this paper and not included in Appendix A can be obtained from the first author, Matthias Elstner, upon request. The proteomic data presented in this study are available in the ProteomeXchange repository under accession number PXD010489.

## 5. Conclusions

Muscle fibers showing bioenergetic deficit (COX-) as a consequence of mtDNA mutations in CPEO upregulate mRNA transcripts involved in translational elongation and protein synthesis.

At the transcriptional level, there are significant differences in mitochondria-, ubiquitination- and cell death-related mRNAs between bioenergetically compensated COX+ and deficient COX- fibers.

Through a hypothesis-free omics approach, we have generated a resource database to further mine the transcriptional features of bioenergetic deficit caused by mtDNA mutations and their relationship to the proteome in skeletal muscle fibers.

## Figures and Tables

**Figure 1 ijms-22-11080-f001:**
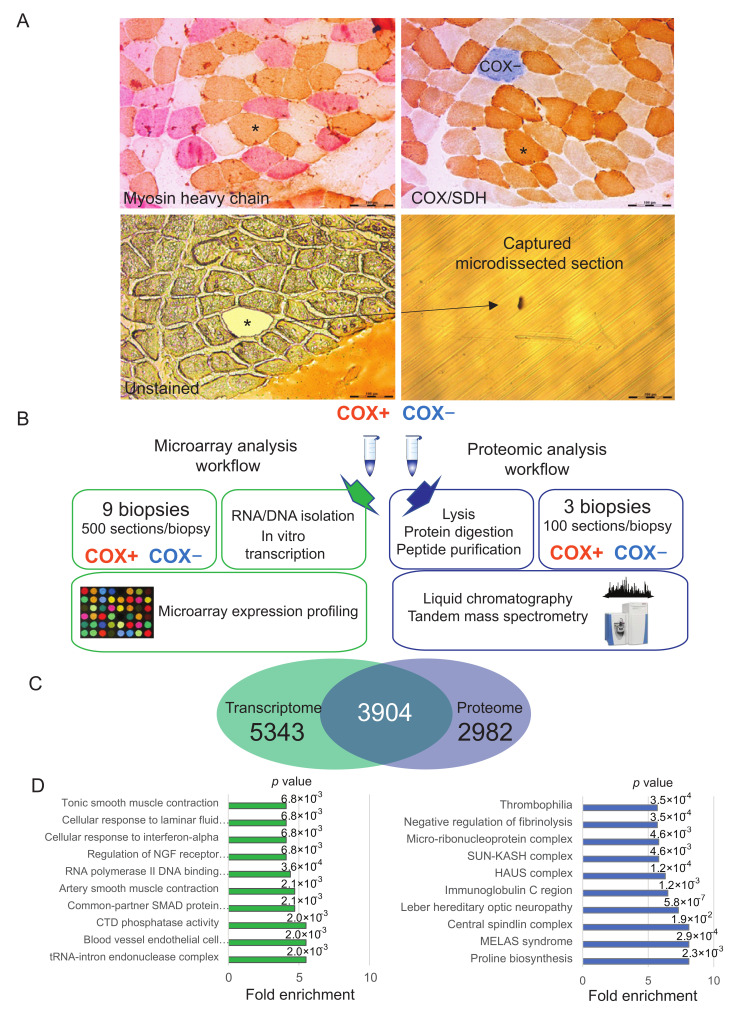
Laser capture microdissection-based workflows and features of the datasets. (**A**) Serial sections of skeletal muscle from a patient suffering from chronic progressive external ophthalmoplegia CPEO. Top left, myosin heavy chain staining for fiber type (brown, type 1-slow fibers; pink, fast-2A fibers.; white, fast-2X fibers). Top right, cytochrome c oxidase (COX)/succinate dehydrogenase (SDH) staining. Bottom left, unstained serial section used for laser microdissection as indicated. The fiber labeled with an asterisk in the serial sections was microdissected, as schmatically depicted Bar, 200 *µ*M. (**B**) Schematic representation of the workflow of laser capture microdissection coupled to both the transcriptome analysis presented here and our previous proteomic dataset. Pools of 500 COX+ and COX- fibers were obtained from 9 patients for transcriptome data and pools of 100 from 3 patients for proteome data. (**C**) Venn diagram of data integration, showing the gene products quantified both at the transcriptome and proteome level and those unique for each dataset. (**D**) Top ten annotations significantly enriched in the gene products quantified only in the transcriptome (left, green) and only in the proteome (right, blue). *p* value is reported on top of each corresponding bar. Fisher exact test, Benjamini-Hochberg FDR = 0.02 for truncation.

**Figure 2 ijms-22-11080-f002:**
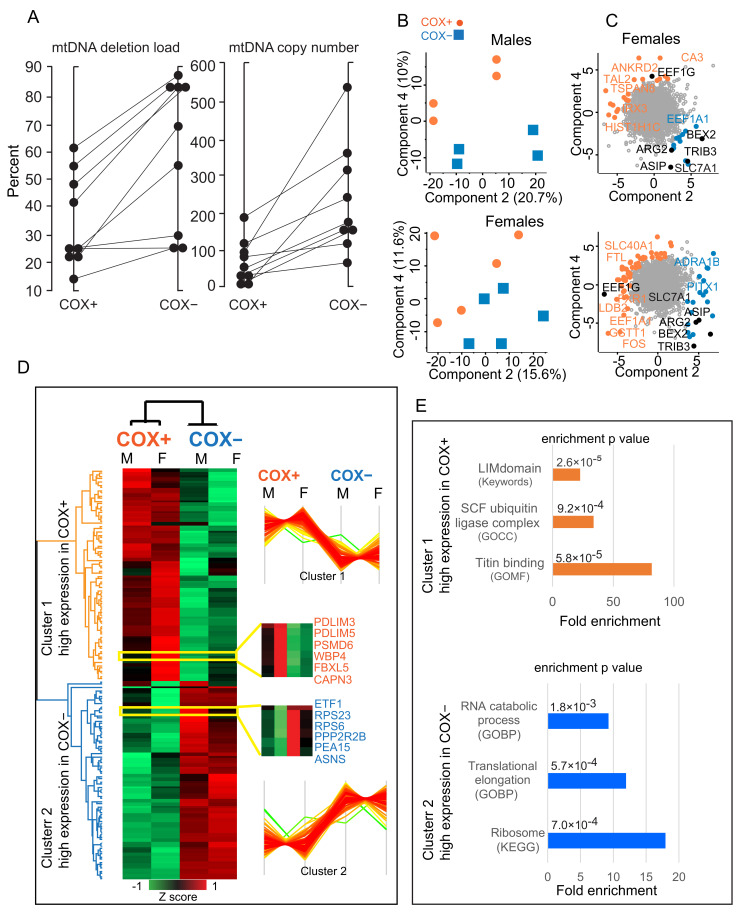
Transcriptomic features of cytochrome oxidase (COX)+ and COX- muscle fibers. (**A**) mtDNA deletion load (%) and copy number (%) of COX-positive (COX+) and COX-negative (COX-) muscle fibers. Each dot represents the value of 500 pooled fiber sections. Fibers isolated from the same biopsy are connected with a line. Average deletion load was COX+ = 35%, COX- = 60% (*p* = 0.002). Copy number increased accordingly (*p* = 0.007). (**B**) Principal component analysis (PCA) components 2. and 4. COX+ fibers are indicated by orange dots, COX- by blue squares. (**C**) PCA loadings, highlighting the transcripts driving the separation into components being expressed at higher level in COX+ (orange) and COX- (blue) fibers. Top, males. Bottom, females. Loadings which are common hits to both graphs are labeled in black. (**D**) Unsupervised hierarchical clusters of 148 transcripts with significantly different expression in COX+ and COX- fibers. For each lane, we calculated the median expression in male and female patients. The colors of the dendrograms of the two main clusters match the colors of the corresponding enrichments in panel E. Data are Z-scored as indicated. Insets show transcripts displaying the expression profile shown on the right of the heatmap, with differences between male and female patients. The Z score level of each cluster is shown on the right of the heatmap. (**E**) Main annotation enrichments in top cluster (higher expression in COX+, orange) and bottom cluster (higher expression in COX-, blue) with corresponding *p* value. Each enriched annotation is labeled with the corresponding database source, i.e., GO, Keywords, and KEGG. Calculated by Fisher exact test, Benjamini-Hochberg FDR = 0.02 for truncation. *n* = 9 patients, 5 females and 4 males.

**Figure 3 ijms-22-11080-f003:**
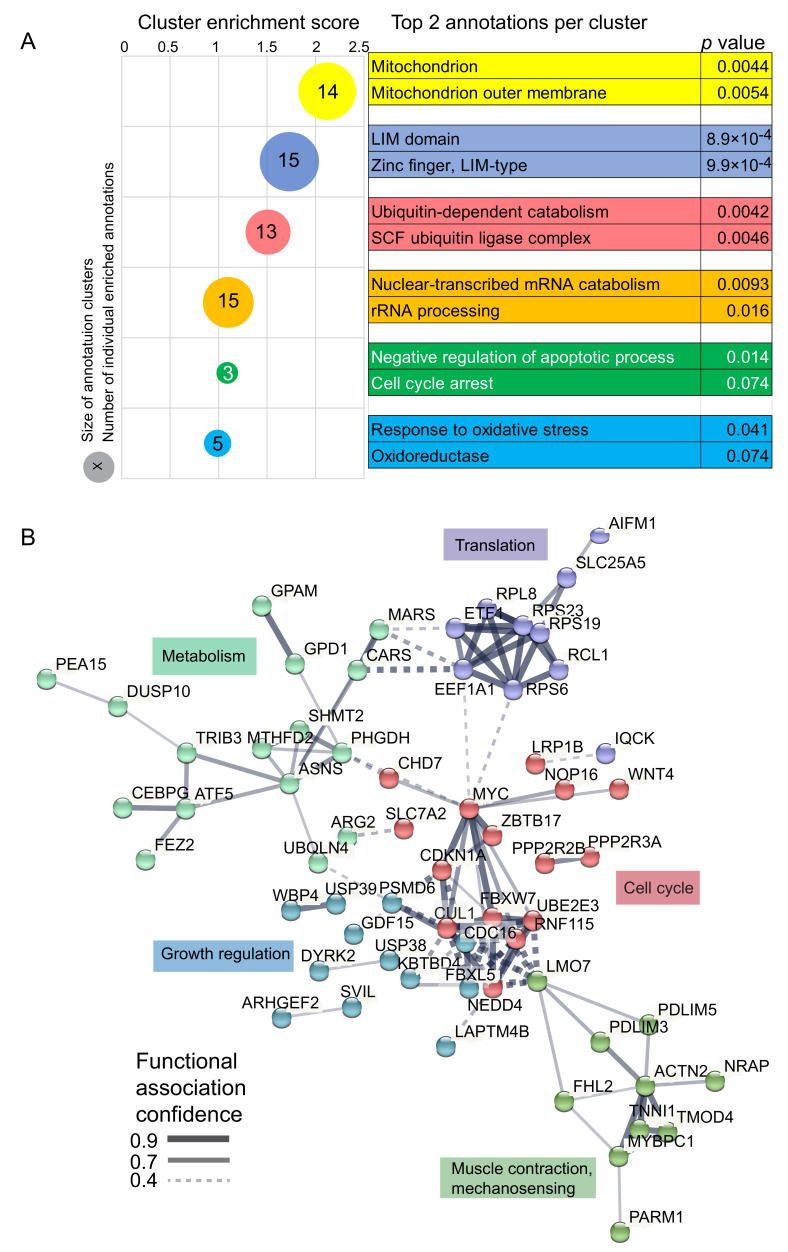
Gene Functional Classification and STRING analysis of transcripts with significantly different expression between cytochrome oxidase (COX)+ and COX- fibers. (**A**) DAVID Gene Functional Classification carried out on 148 transcripts with significantly different expression between COX+ and COX- fibers of 9 patients. Six clusters with enrichment >1 were highlighted at medium stringency. The bubble plot shows fold enrichment on the x axis. Colors match the legend on the right. The size of the bubble is proportional to the numbers of individual enriched annotations within the main clusters. The number reports the significant individual hits in each cluster. (**B**) STRING functional interaction network analysis of statistically significant gene products. In this protein-centric view, nodes show functionally interacting gene products, edges represent the confidence of functional interaction (see the legend bottom left). We set the analysis to highlight five clusters, distinguished by node color, enriched in the biological function displayed in color next to each cluster (curated).

**Figure 4 ijms-22-11080-f004:**
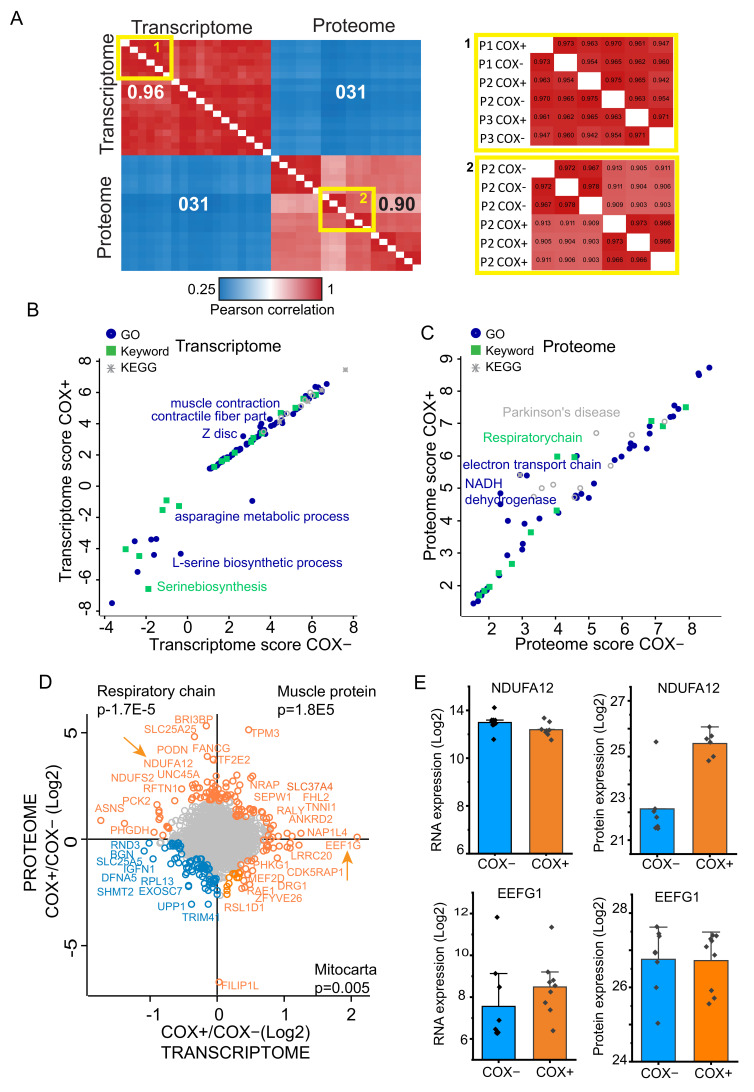
Integration of dataset and multi-omics analysis of common gene products. (**A**) Multi-scatter plot-based heatmap of Pearson correlations among all transcriptome and proteome single run. *n* = 9 patients for transcriptome. *n* = 3 patients for proteome. Pearson correlation color scale is displayed at bottom. Insets on the right show enlarged areas of the main heatmap (areas with yellow borders). (**B**) Scatter plot of 2D annotation enrichments comparing cytochrome oxidase (COX)+ and COX- fibers at the transcriptome level (*n* = 9 patients). Functional annotations (GO, Keywords, and KEGG) are labeled with different shapes and color as indicated in the legend top left. (**C**) Scatter plot of 2D annotation enrichments comparing COX+ and COX- fiber at the proteome level (*n* = 3 patients). Legend, see panel B. (**D**) Scatter plot of Log2 COX+/COX- ratio directly comparing transcriptome and proteome features. Top right quadrant: higher expression in COX+ at both transcriptome and proteome level. Top left quadrant, high in COX+ at the proteome level. Bottom right quadrant, high expression in COX+ at the transcriptome level. Top enrichment with *p* values is shown for the three quadrants. Scatter plot of Log2 COX+/COX- ratio directly comparing transcriptome and proteome features. Top right quadrant: higher expression in COX+ at both transcriptome and proteome level. Top left quadrant, high in COX+ at the proteome level. Bottom right quadrant, high in COX+ at the transcriptome level. Top enrichment with *p* values is shown for the three quadrants. Two arrows indicate gene products analyzed in more detail. (**E**) Bar graph of Log_2_ individual data points comparing the RNA (left column) and protein expression (right column) of NDUFA12 and EEFG1, two gene products indicated by an arrow in panel D. Bars show the median expression, whiskers the standard deviation. *n* = 9 patients for transcriptome. *n* = 3 patients, each in technical triplicates, for proteome.

## Data Availability

Transcriptomic data not included in Appendix A are available upon request to ME.

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
