# Peer review of "Multi-Omics Approach to Mitochondrial DNA Damage in Human Muscle Fibers"

_ijms, 2021, doi:10.3390/ijms222011080_

Round 1

Reviewer 1 Report

The manuscript of Elstner et al. reports transcriptome and proteome analysis of COX- and COX+ muscle fibres from CPEO patients. While muscle fibres were histochemically characterized, microdissection allow omic data generation.

The authors then depicted the differences upon COX status to identify potential gene/pathways involved in the disease-related phenotype.

The manuscript could be improved, especially the presentation better clarity.

  • Lines 42-45 : please provide references supporting the origin of mtDNA deletion
  • Lines 53-56 : please provide references showing that mtDNA deletions encompass at least one mt-COX subunit, and thus affect COX activity.
  • Line 71-72 : Please provide corresponding references, and report major findings.
  • Line 76 : Please clarify what is the reference for "the adaptative response": to mtDNA deletion, to COX deficiency, to muscle disease,...??
  • Line 95 (Figure 1) and lines 101-117: Some elements of the figure could be moved to the M&M section, while some important information are missing or unclear for the reader. It is not clearly stated if and how the pools are coming from a single sample of a patient. How many pools were made and analyzed, from which patient ? These data could also be included in Table S1.
  • General remark : all along the manuscript it is not easy to determine the number of replicates/analysis that led to the presented results
  • Line 139 : There is no information about data repository for omics
  • Line 139 : The 148 identifed regulated genes are not mentioned
  • Line 151 Figure 2 and lines 170-177 : it is difficult to understand the separation orange/blue in panel E. GO enrichments and gene overexpression should be distinguished are better presented. Please clarify the whole §
  • Lines 247-254: Same as above, clarify annotations and expression.
  • lines 286 (Figure 4) : There is no clear indication of the number of experiments that generated the results. Neither in graphs or legends, it is not possible to understand if results are replicate or individual experiments. In panel E, for protein determination of NDUFA12, there are 4 dots while for EEFG1 there are more than 10. Please explain and clarify.
  • Line330 : The 148 significantly regulated genes could have been analyzed in detailled and discussed in regards to the previous published omics.
  • Lines 349- Materials and Methods : This section could be improve giving more details and providing missing information. Example line 380 :  the mtDNA heteroplasmy was determined on microdissected fibres or whole muscle tissue ? No indication of repository for transcriptome and proteome data. No description of statistical analysis.
  • Table S2 : no indication of statistics : column U (and column X ??) only indicates p value, no description of adjustements if any ? No column with fold changes, but many redondant columns. The table should be redesign to improve clarity.

Reviewer 2 Report

Pleas enumerate the major findings of the study in the abstract 

Please inform the number of patients in the abstract 

please provide a brief (general) description of proteome and transcriptome  in the introduction section 

 Line 43 and 45: Please provide reference (PPR) 

Line 50:  PPR

Line 51-58 PPR

Line 61: PPR 

line 66 : PLease describe   breifly OXPHOS deficiency 

Line 76: PPR

Line 80 -88 Please rephrase mainly the primary objective of the study 

Line 93-104 , is some word missing? Please finish a paragraph before inserting a figure to avoid  misinterpretation 

Line 230 please elaborate possible post transcriptional mechanism 

Line 269-272 please develop this paragraph 

Line  345 

This conclusion is poor please provide a conclusion or suggestion directly in correlation with the findings of this study

Fig 3 is difficult to read please provide a better resolution 

Was the main objective of the study achieved 

Line 351  Please rephrase as it is confusing (was it a written informed study from patients ? )

Line 402 PPR

for conclusion 

I think a  brief summary enumerating major findings including the primary objective would be helpful 

Reviewer 3 Report

Elstner and colleaques analyzed the mosaic muscle phenotype of CPEO patients characterized by fibers harbouring wildtype (CO+ fibers) or mutated mtDNA (CO- fibers) in the same muscle using an omics technologies. Overall the study is well conducted and well written. Neverthless, few changes can further improve the quality of the study:

  • line 94 seems to be incomplete 
  • the panel of the unstained field represented in figure 1 seems "too distant"/not "next section" of the one in the panel of COX staining
  • It is not mentioned in the text, nor in table S1 the name of the muscle(s) analyzed in the study
  • mtDNA mutations arise in post-mitotic tissue, however skeletal muscle in aging and in other degenerative conditions can frequently present regenerating fibers characterized by the presence of centrally-located nusclei. Have the samples been checked/excluded for the presence of regenerating fibers? A panel representing the quality of sample by H&E staining (showing the position of myonuclei) can be added in Figure 1 and described in the text, since it also affetcs the mitochondrial biology.

Round 2

Reviewer 1 Report

The authors have made extensive work to answers all the comments raised, and have deeply improved the manuscript. 

The manuscript can thus be published in its present version.

Reviewer 2 Report

The authors have adequately responded to all queries and I congratulate them